# ATF3-Expressing Large-Diameter Sensory Afferents at Acute Stage as Bio-Signatures of Persistent Pain Associated with Lumbar Radiculopathy

**DOI:** 10.3390/cells10050992

**Published:** 2021-04-23

**Authors:** Jiann-Her Lin, Yu-Wen Yu, Yu-Chia Chuang, Cheng-Han Lee, Chih-Cheng Chen

**Affiliations:** 1Division of Neurosurgery, Department of Surgery, Taipei Medical University Hospital, Taipei 110301, Taiwan; jiannher@tmu.edu.tw; 2Department of Surgery, School of Medicine, College of Medicine, Taipei Medical University, Taipei 110301, Taiwan; yvonneyu@ibms.sinica.edu.tw; 3Taipei Neuroscience Institute, Taipei Medical University, Taipei 110301, Taiwan; 4Institute of Biomedical Sciences, Academia Sinica, Taipei 115201, Taiwan; ycchuang@ibms.sinica.edu.tw (Y.-C.C.); hans@ibms.sinica.edu.tw (C.-H.L.); 5Neuroscience Program of Academia Sinica, Academia Sinica, Taipei 115201, Taiwan; 6Taiwan Mouse Clinic, Biomedical Translation Research Center, Academia Sinica, Taipei 115202, Taiwan

**Keywords:** ATF3, lumbar radiculopathy, nerve constriction, neuropathic pain, chronic pain, dorsal root ganglion, neurofilament, hypoxia

## Abstract

The mechanism of pain chronicity is largely unknown in lumbar radiculopathy (LR). The anatomical location of nerve injury is one of the important factors associated with pain chronicity of LR. Accumulating evidence has shown constriction distal to the dorsal root ganglion (DRG) caused more severe radiculopathy than constriction proximal to the DRG; thereby, the mechanism of pain chronicity in LR could be revealed by comparing the differences in pathological changes of DRGs between nerve constriction distal and proximal to the DRG. Here, we used 2 rat models of LR with nerve constriction distal or proximal to the DRG to probe how the different nerve injury sites could differentially affect pain chronicity and the pathological changes of DRG neuron subpopulations. As expected, rats with nerve constriction distal to the DRG showed more persistent pain behaviors than those with nerve constriction proximal to the DRG in 50% paw withdraw threshold, weight-bearing test, and acetone test. One day after the operation, distal and proximal nerve constriction showed differential pathological changes of DRG. The ratios of activating transcription factor3 (ATF3)-positive DRG neurons were significantly higher in rats with nerve constriction distal to DRG than those with nerve constriction proximal to DRG. In subpopulation analysis, the ratios of ATF3-immunoreactivity (IR) in neurofilament heavy chain (NFH)-positive DRG neurons significantly increased in distal nerve constriction compared to proximal nerve constriction; although, both distal and proximal nerve constriction presented increased ratios of ATF3-IR in calcitonin gene-related peptide (CGRP)-positive DRG neurons. Moreover, the nerve constriction proximal to DRG caused more hypoxia than did that distal to DRG. Together, ATF3 expression in NHF-positive DRG neurons at the acute stage is a potential bio-signature of persistent pain in rat models of LR.

## 1. Introduction

The mechanism of pain chronicity is largely unknown in lumbar radiculopathy (LR). LR, defined as radiating leg pain below the knee with neurological deficits in the distribution of the lumbosacral nerves, is an important clinical problem that can lead to chronic and debilitating pain [1]. The majority of LR spontaneously resolved after 3 months [2], but once pain chronicity was established, treatment was not effective [3]. Accordingly, it is important to identify the patients who will develop chronic pain in an early phase. A nationwide cohort showed 40% of severe LR developed as chronic pain after 13 years and suggested the severity of LR is associated with pain chronicity [4]. Substantial evidence showed different anatomical locations of lumbosacral nerve compression have impacts on the severity of LR. A prospective cohort of 75 patients revealed that, compared to central lumbar canal stenosis, lateral lumbar canal stenosis is associated with more severe symptoms and more sensory deficits in patients with LR [5]. Evidence from animal models of LR also demonstrated animals perform dramatically different disease severity that largely depends on the constriction sites in relation to the dorsal root ganglion (DRG). Animals with the constriction sites distal to the DRG presented more sensitive mechanical hyperalgesia than those with constriction sites proximal to the DRG [6,7]. In an electrophysiological study, compound actin potential of sensory nerves was significantly decreased 2 weeks after the injury in animals with the constriction sites distal to the DRG but not in those with constriction proximal to DRG [8]. The frequency and amplitude of the excitatory post-synaptic current in the dorsal horn were significantly increased in animals with the constriction distal to the DRG but not in those with the constriction proximal to the DRG [6]. Animals with the constriction distal to the DRG had significantly more microglia activation [6,9] on the 7th day after injury in the dorsal horn compared to those with the constriction proximal to the DRG. In DRGs, the expression of tumor necrosis factor-alpha (TNF-α) and the number of apoptotic neurons were significantly higher in distal constriction animals than in proximal constriction animals [7]. Thereby, the anatomical location of nerve injury is associated with severity and pain chronicity of LR.

Although accumulating evidence has shown constriction distal to the DRG caused more severe radiculopathy than constriction proximal to the DRG, the pathophysiology is largely unknown. One hypothesis is that the severity of radiculopathy is associated with the extent of ischemia at the DRG based on the constriction’s spatial relation to the DRG. Root constriction has been shown to largely affect the blood flow of nerve roots [10]. The blood flow rate of the nerve root is decreased by 69% when the nerve root is constricted distal to the DRG as compared with the blood flow by 37% when proximal to the DRG [11]. Thus, the more distal the constriction of the nerve root occurs, the more ischemic changes are induced. Another hypothesis is large-diameter sensory afferents injury. Several studies showed that large-diameter sensory afferents injury is associated with the severity of the peripheral nerve injury [5,12]. The expression of calcitonin gene-related peptide (CGRP) in large-diameter sensory afferents was associated with positive neuropathic syndrome [12]. In patients with LR, impaired function of large-diameter sensory afferents (dermatomal sensory defect) was associated with more severe symptoms [5]. We believed that the mechanism of pain chronicity in LR could be revealed by highlighting the differences of pathological changes of DRGs between nerve constriction distal and proximal to the DRG at the acute stage, and, for the purpose of prevention, these differences could serve as a bio-signature in patients with LR. Thereafter, we aim (1) to compare the pain phenotypes following nerve constriction proximal or distal to the DRG and (2) to investigate the tissue hypoxia of DRG and ATF3 expression among DRG neuron subpopulations, especially the large-diameter sensory afferent, in these 2 LR rat models.

## 2. Methods

### 2.1. Animals Handling and Preparation

Adult male Sprague–Dawley (SD) rats (250–300 g) were obtained from the National Laboratory Animal Center, Taipei, Taiwan. The rats were provided food and water and were maintained on a 12 h light/dark cycle in a temperature- and humidity-controlled animal center at both medical institutions.

All experimental protocols were performed in accordance with the Guidelines for Animal Experiments of Taipei Medical University and Academia Sinica, and the Guiding Principles for the Care and Use of Laboratory Animals approved by the Chinese Society of Laboratory Animal Sciences, Taiwan (LAC-100-0221 and 16-12-1022). A minimal number of rats were used for each study, and all efforts were made to minimize potential suffering.

### 2.2. Experimental Design

In the behavioral test, the rats were divided into four groups (N = 6/each group): control (naïve), sham, proximal nerve constriction, and distal nerve constriction. For quantitative image analysis of L5 DRG neurons, the rats were divided into three groups (N = 6/each group): sham, proximal nerve constriction, and distal nerve constriction. The operation procedures are described below. The spinal nerve 2 mm proximal to the DRG were constricted by 4-0 silk in the proximal nerve constriction group, whereas 2 mm distal to the DRG in the distal nerve constriction group. The sham group received all the operation procedures but no constriction on the spinal nerve, and the control group received anesthesia but no operations.

### 2.3. Operation

To probe the molecular mechanism and therapeutic targets of LR, we used two animal models of LR—(1) Proximal nerve constriction: constriction 2 mm proximal to the DRG; (2) distal nerve constriction: constriction 2 mm distal to the DRG, as described by Winkelstein et al. [13]. LR is mostly resulted from compression of the lumbosacral nerves due to disc herniation or bony structure stenosis at different anatomical locations within the spinal canal [14,15]. For the L5 spinal nerve, the pathology at the central lumbar spinal canal comprises the dorsal root, which is proximal to the dorsal root ganglia (DRG), whereas the pathology at the foramen (lateral lumbar spinal canal) comprises the DRG or the spinal nerve just distal to the DRG [16,17,18]. The animals were anesthetized with ketamine (80–200 mg/kg) through intraperitoneal injection. The back was sterilized with iodine and alcohol. The L5–L6 intervertebral space was identified by palpation, and a midline incision was made from L3 to L6. The paraspinal muscles were dissected free from the spinal processes on one side. The transverse processes of L5 and L6 were exposed by scraping off attached ligaments, and a laminectomy was performed to expose an approximately 1 cm length of the spinal cord. Both the right pedicles of L5 and L6 were removed by drilling to expose the right L5 DRG. The dura was opened over 3–5 mm in length, and the right L5 dorsal root and its DRG were identified under a dissecting microscope. A 4-0 silk constriction was placed on 2 mm proximal or distal to the DRG of the spinal nerve. Then the tissues, muscles, and skin were closed with silk. (Figure 1).

### 2.4. 50% Paw Withdraw Threshold (PWT)

All prospective mechanosensitivity experiments with animals were conducted in accordance with the guidelines from the Taiwan Council on Animal Care and the approval of the Taipei Medical University Animal Care Committee. 50% PWT was measured in animals according to the simplified up-down method [19]. Rats were placed in acrylic chambers (12.5 cm × 20 cm × 15 cm) suspended above a wire mesh grid and allowed to acclimatize to the testing apparatus for 1 h prior to experiments. When the rat was not moving, the von Frey filaments were pressed against the plantar surface of the paw until the filament buckled and held for a maximum of 3 s. A positive response was determined if the paw was sharply withdrawn on the application of the filament or a paw flinching immediately upon the removal of the filament.

### 2.5. Weight-bearing Test

Changes in restricted weight distribution between the left and right hind limbs were determined using an incapacitance tester (Columbus Instruments, Columbus, OH, USA). The rats were placed in an angled Plexiglas chamber so that each hind limb was positioned on a separate force plate. The rats were allowed to acclimate to the apparatus, and when stationary, the weight distribution readings were taken. The downward force (measured in grams) applied by each hind limb was assessed and averaged over a three-second period. The measurement was performed five times on each rat, and the mean of the middle three values was calculated. The difference of weight distribution between the left and right (operated side) hind paw was calculated. For example, the percent weight distribution of the right hind paw was calculated by the following formula [20]: % weight distribution of left hind paw = right weight/(left weight + right weight) ×100%

### 2.6. Acetone Test

Cold sensitivity was assessed using the acetone drop method, as described by Choi et al., with modification [21]. The rats were placed in a metal mesh cage and allowed to habituate for approximately 30 min. Freshly dispensed acetone drop (50 μL) was applied gently onto the mid plantar surface of the hind paw. The responses upon the application of acetone were ignored. Typically, 2–5 s after application, a cold-sensitive reaction, with respect to either paw licking, shaking or rubbing the hind paw, and brisk foot withdrawal was recorded as a positive response. The responses were measured for one minute. The interval between each application of acetone was approximately 5 min. For each measurement, the right paw was sampled in three trials, and an acetone test score was given to each trial. Each trial was scored by the following scoring system:

0: no positive response

1: <3 positive response

2: 3–4 positive response

3: >5 positive response

4: continuous positive responses until licking

The final acetone test score of the right paw was the sum of the acetone test scores of all three trials.

### 2.7. Immunohistochemistry

For evaluation of tissue hypoxia, the rats were intraperitoneally injected with hypoxyprobe-1 (HO-1, 60 mg/kg pimonidazole HCl, Hypoxyprobe, Inc., Burlington, MA, USA). After 1 h, isoflurane-anesthetized rats were transcardially perfused with 100 mL of ice-cold normal saline, followed by 100 mL of ice-cold 4% *w/v* paraformaldehyde (PFA, Sigma-Aldrich, Darmstadt, Germany) in 0.1 M phosphate-buffered saline (PBS), and performed a laminectomy at postoperative day 1. The L5 DRGs from rats were dissected then dehydrated with 30% sucrose overnight at 4 °C then were embedded in Tissue-Tek OCT. The tissue was cut in a Leica cryostat at 12 μm thickness storing at −20 °C. After three washes with PBS, the slices were blocked for 1 h at room temperature in PBS, containing 5% bovine serum albumin (BSA) and 0.1% Tween 20, then incubated overnight with the designated primary antibodies reconstituted in blocking buffer at 4 °C. Primary antibodies or fluorescence-conjugated agents used in this study were rabbit anti-ATF3 (1:1000, Santa Cruz Biotechnology, Santa Cruz, CA, USA), mouse anti-NeuN (1:1000, Abcam, Cambridge, MA, USA), chicken anti-NFH (1:1000, Abcam, Cambridge, MA, USA), goat anti-CGRP (1:1000, Serotec, Kidlington, UK), FITC-conjugated Isolectin B4 (IB4, Sigma-Aldrich, Milan, Italy), and FITC-conjugated mouse anti-HO-1 (Hypoxyprobe™). Sections were rinsed then incubated with a secondary fluorescent antibody Alexa Fluor 488 goat anti-rabbit or 594 donkey anti-rabbit antibody, 647 goat anti-mouse or 647 donkey anti-mouse antibody, 488 goat anti-chicken antibody, and 488 donkey anti-goat antibody (1:1000, Invitrogen technologies, Carlsbad, CA, USA). DRG sections were also counterstained with 4′,6-diamidino-2-phenylindole (DAPI) for nuclear labeling before mounting. For IB4 staining, sections were incubated with 4 μg/mL FITC-conjugated IB4 in 0.1 M PBS containing 0.1 mM CaCl_2_, 0.1 mM MgCl_2_, and 0.1 mM MnCl_2_ at room temperature for 90 min, then rinsed and mounted with 0.1 M PBS. Finally, all sections were cover-slipped under a VECTASHIELD^®®^ antifade mounting medium with nail polish. Images of DRG sections were acquired under an LSM 700 confocal microscope (Carl Zeiss, Oberkochenm, Germany) and processed with Zen 2.3 software.

### 2.8. Data Sampling and Analysis

For pain behaviors, two-way ANOVAs were used to test the difference between groups and between time points, and post hoc Tukey’s multiple comparison tests were used to test the significant difference between each time point. Prism7 for Mac OS X (GraphPad Software, Inc., San Diego, CA, USA) was used for statistical analysis.

For quantitative image analysis of the L5 DRG neurons, DRG cryosections were cut 12-μm–thick, and 6 sections were sampled in a slide to represent a ganglion, by which each section was collected in every 96 μm (or more). After staining, images were digitized using the Pannoramic 250 Flash II (3DHISTECH), equipped with a 20× objective. Subsequently, images were processed with ImageJ 64, and a blinded experimenter manually counted each different immunoreactive profile of neuronal markers with visible nuclei. The IR of NeuN was used as a pan-neuronal marker while NFH, CGRP, and IB4 as subtype neuronal markers. The ratios of co-localization of HO-1 or ATF3 with neuronal markers were calculated. All statistical analyses were calculated by GraphPad Prism version 7.0. A one-way ANOVA with post hoc Holm–Sidak multiple comparisons were performed the difference between groups.

## 3. Results

### 3.1. Animals with Distal Nerve Constriction Presented More Persistent Pain Behaviors Than Those with Proximal Nerve Constriction

Previous studies have shown that rats with nerve injury distal to the DRG had more severe mechanical hyperalgesia than those with nerve injury proximal to the DRG during the 4 weeks after the operation [6,7]. However, no information was available about pain behaviors in a longer time course or other pain behaviors modalities. We, therefore, examined the differential effects of nerve injury distal or proximal to the L5 DRG on rat pain phenotypes for 12 weeks.

In the von Frey test (reflecting to mechanical hyperalgesia or allodynia), two-way ANOVA analysis revealed a significant difference in the 50% PWT between groups: time F(13, 260)= 11.64, *p* < 0.0001; treatment F(3, 20) = 41.3, *p* < 0.0001; interaction F(39, 260) = 1.718, *p* = 0.0075 (Figure 2A). The post hoc test showed, compared with the control group, animals with distal nerve constriction presented significantly lower PWT from the 4th day to the 12th week (Tukey’s multiple comparison test, *p* < 0.001 at each time point). In contrast, those with proximal nerve constriction presented significantly lower PWT only from the 4th day to the 7th week (Tukey’s multiple comparison test, *p* < 0.01 at each time point), and the sham group only showed significantly lower PWT from the 4th day to the 1st week (Tukey’s multiple comparison test, *p* < 0.01 at each time point). In a head-to-head comparison between animals with distal nerve constriction and those with proximal nerve constriction, there was no significant difference before 9th week; however, those with distal nerve constriction showed significantly lower threshold from the 10th to 12th week (Tukey’s multiple comparison test, *p* < 0.05 at each time point).

In the acetone tests (reflecting cold allodynia), two-way ANOVA analysis revealed a significant difference between groups: time F(13, 260) = 2.343, *p* < 0.0001; treatment F(3, 20) = 25.44, *p* < 0.0001; interaction F(39, 260) = 2.343, *p* < 0.0001 (Figure 2B). The post hoc tests showed, compared with the control group, animals with distal nerve constriction presented significantly higher acetone-test scores for 11 weeks (Tukey’s multiple comparison test, *p* < 0.05 at each time point). In contrast, those with proximal nerve constriction and sham surgery presented significantly higher acetone-test scores before the 4th week and before the 2nd week, respectively (Tukey’s multiple comparison test, *p* < 0.05 at each time point). In a head-to-head comparison between nerve injury groups, those with distal nerve constriction showed significantly higher acetone-test scores after 3 weeks as compared with those with proximal nerve constriction (Tukey’s multiple comparison test, *p* < 0.05 at each time point).

In the weight-bearing test (reflecting the guarding pain), two-way ANOVA analysis revealed a significant difference between groups: time F(13, 260) = 24.03, *p* < 0.0001; treatment F(3, 20) = 92.5, *p* < 0.0001; interaction F(39, 260) = 6.065, *p* < 0.0001 (Figure 2C). The post hoc test showed animals with distal nerve constriction had significantly higher weight-bearing than the control group starting at 4 days after the operation and lasting for 12 weeks (Tukey’s multiple comparison test, *p* < 0.0001 at each time point). In contrast, animals with proximal nerve constriction had significantly higher weight-bearing than the control group starting at 4 days after the operation but only lasting for 4 weeks (Tukey’s multiple comparison test, *p* < 0.01 between the 4th day and 4th week). The sham group also showed a significant difference as compared with the control group but only lasting for 1 week. In a head-to-head comparison between animals with distal nerve constriction and those with proximal nerve constriction, the difference began at the 3rd week, and these differences persisted until the end of the test (Tukey’s multiple comparison test, *p* < 0.05 at 3rd, 4th, and 5th week; *p* < 0.0001 after 6th week until the end of the test).

In summary, all three pain behavioral tests demonstrated that neuropathic pain phenotypes lasted longer in animals with distal nerve constriction than those with proximal nerve constriction.

### 3.2. More Severe Hypoxia Occurred in the DRGs after Nerve Constriction Proximal to DRG Than Those Distal to DRG

HO-1 is an exogenously administered probe that is activated under conditions of hypoxia (pO_2_ < 10 mmHg) [22], creating thiol attaching on proteins that can then be examined by immunohistochemical techniques. Since the blood flow of DRGs is reduced more in constriction distal to the DRG than in constriction proximal to the DRG [11], more severe hypoxia should be present in the DRGs after distal nerve constriction. We, therefore, compared the extent of expression of HO-1 in DRGs 1 day after distal nerve constriction with those after proximal nerve constriction. The expression of HO-1 after proximal nerve constriction was significantly higher than that after distal nerve constriction and even those in the sham group (one-way ANOVA, *p* = 0.0026, Distal vs. Proximal vs. Sham, 17.86 ± 5.14 vs. 40.73 ± 7.05 vs. 11.47 ± 1.47%, respectively) (Figure 3). The results of hypoxia in the present study were not consistent with the blood flow direction hypothesis.

### 3.3. The Ratios of ATF3-Positive DRG Neurons Increased Significantly After Nerve Constriction Distal to DRG Than After Nerve Constriction Proximal to DRG

Next, we tested whether the more severe DRG hypoxia due to proximal nerve constriction would result in more neuron damage compared to distal nerve constriction. Therefore, we compared the ratios of ATF3-positive DRG neurons between distal nerve constriction, proximal nerve constriction, and sham groups. In L5 DRGs, the ratio of ATF3-positive DRG neurons was significantly higher in the distal nerve constriction group than that in the proximal nerve constriction group, and the latter was significantly higher than that in the sham group at postoperative day 1 (distal vs. proximal vs. sham, 39.81 ± 2.41 vs. 23.78 ± 2.96 vs. 10.74 ± 2.17%, respectively) (Figure 4). We further analyzed ATF3 expression in different subsets of DRG neurons. In NFH-positive subpopulations, the ratio of ATF3-positive neurons in the distal constriction group was significantly higher than that in the proximal nerve constriction or sham groups, but there was no significant difference between the proximal nerve constriction and sham groups (one-way ANOVA, *p* = 0.0002, distal vs. proximal vs. sham, 57.37 ± 4.54 vs. 33.90 ± 4.80 vs. 25.55 ± 2.40%, respectively). In IB4-positive subpopulations, there was no difference among groups in the ratios of ATF3-positive neurons (one-way ANOVA, *p* = 0.0542, distal vs. proximal vs. sham, 30.91 ± 3.84 vs. 17.44 ± 3.36 vs. 17.55 ± 4.95%, respectively). In CGRP-positive subpopulations, the ratio of ATF3-positive neurons in the distal or proximal nerve constriction group was significantly higher than that in the sham group, but there was no significant difference between the distal and proximal nerve constriction groups (one-way ANOVA, *p* = 0.0225, distal vs. proximal vs. sham, 30.77 ± 3.20 vs. 32.09 ± 4.92 vs. 17.29 ± 2.53%, respectively).

### 3.4. The Ratios of CGRP-Positive DRG Neurons Increased 1 Day After Proximal Nerve Constriction

Finally, we examined the impact of different spinal nerve injuries on the expression of NFH, CGRP, and IB4-binding signals in DRG neurons 1 day after nerve constriction. The ratios of NFH-positive (one-way ANOVA, *p* = 0.8871) or IB4-positive (one-way ANOVA, *p* = 0.5993) DRG neurons were not different between groups, but the ratios of CGRP-positive DRG neurons were significantly different among the three groups (one-way ANOVA, *p* = 0.0486) (Figure 5). The ratio of CGRP-positive neurons in the proximal constriction group was significantly higher than that in the sham group (distal vs. proximal vs. sham, 29.08 ± 1.79 vs. 32.46 ± 1.55 vs. 24.95 ± 2.40%, respectively).

## 4. Discussion

In this study, we have successfully used proximal or distal nerve constriction of the L5 DRG to mimic LR associated with central or lateral lumbar spinal canal stenosis, respectively. Consistent with the clinical observation, chronic constriction of the spinal nerve either proximal or distal to the DRG caused unilateral hyperalgesia and allodynia in the rats’ hind paws, and animals with nerve constriction distal to the DRG had more persistent pain behaviors than those with nerve constriction proximal to the DRG. On the first day of nerve constriction, animals with distal nerve constriction showed higher ratios of ATF3-expressing DRG neurons as compared with those with proximal nerve constriction, especially in the NFH-expressing subpopulation. However, animals with proximal nerve constriction showed more severe DRG hypoxia as compared with those with distal nerve constriction. Thus, the cellular evidence supported the hypothesis of large-diameter sensory afferent injury instead of the blood flow direction hypothesis. This study suggested that ATF3-expressing NFH DRG neurons were associated with pain chronicity of LR.

### 4.1. Possible Pathophysiology Underlying the Different Pain Phenotypes between Nerve Constriction Distal and Proximal to DRG

There are several possible hypotheses explaining the more severe pain behavior after nerve constriction distal to the DRG than after nerve constriction proximal to the DRG. A possible hypothesis for the different severity of radiculopathy is the information flow [6]. The direction of information flow from the periphery to the DRG to the spinal cord itself is a major factor in the distal lesion, giving rise to stronger neuropathic signs. After a DRG injury, both injured and uninjured DRG neurons become excitable and exhibit ectopic firing [23]. The proximal nerve ligation could form a physical barrier to prevent the ectopic firing from reaching the dorsal horn of the spinal cord, while that does not occur in the distal ligation. Another possible hypothesis is peripheral neurotrophic factors. The nerve growth factor from peripheral tissues plays an important role in the survival and maintenance of DRG neurons [24]. The apoptosis of DRG neurons was found to be associated with radiculopathy after peripheral nerve injury [7]. The physical barrier by distal nerve ligation may prevent the signaling of peripheral neurotrophic factors to DRG neurons and, consequently, results in more apoptosis of DRG neurons than proximal nerve ligation. The blood flow direction is one of the most attractive hypotheses because hypoxia after decreased blood flow is an important factor triggering neuropathic pain. The dramatic decrease of blood flow when clipping distal to DRGs may be a consequence of the block of the distal radicular arterial flow or the impairment of the distal radicular venous drainage or both. A proximal radicular artery gives blood flow in a distal direction from the cord towards the spinal nerve, and concomitantly, a distal radicular artery provided blood flow in a proximal direction toward the cord. Although both the proximal and distal radicular arteries give blood supply to DRGs, the blood supply from the distal radicular artery is larger than that from the proximal. It is supported by an experiment that showed there was more decrease of the blood flow in DRG when the distal spinal nerve was clipped than when the proximal spinal nerve was clipped in swine [11]. Hypoxia secondary to compromised blood flow is thought to be one of the possible underlying mechanisms that triggers neuropathic pain associated with LR [25]. Hypoxia induces the apoptosis of DRG neurons in vitro [26,27]. In vivo, hypoxia excites DRG neurons and decreases their firing thresholds in response to mechanical stimuli [28]. Thus, hypoxia might be an important factor to trigger the severe pain phenotypes in animals who received nerve constriction distal to L5 DRG. However, the results of this study showed only animals with proximal nerve constriction showed a significantly higher ratio of hypoxic DRG neurons than sham animals. This finding did not support the blood flow direction hypothesis. The vascular anatomy in spinal nerves of rats may be different from that of swine. Our result suggested that nerve constriction 2 mm distal to the DRG did not impair the blood flow from the distal radicular artery, hinting that the distal radicular artery may enter the L5 spinal nerve at the segment between the nerve constriction site and the DRG in rats. Another explanation is that the blood flow of the L5 DRG in rats is totally supplied by the proximal radicular artery.

Another novel hypothesis is large-diameter sensory afferent injury. Several studies showed that the involvement of large-diameter sensory afferents is associated with the severity of peripheral nerve injury [5,12]. The expression of CGRP in large-diameter sensory afferents was associated with the positive neuropathic syndrome [12]. Plastic changes in the electrophysiological property of large-diameter sensory afferents were demonstrated under pathological pain status [29]. A prospective cohort of 75 patients with LR revealed that defect of sensations conducted by large-diameter sensory afferents was associated with more severe symptoms [5]. This study indicated more ATF3-expressing DRG neurons, especially NFH subpopulation, in distal nerve constriction and supported that large-diameter sensory afferent injury contributes to pain chronicity in distal nerve constriction as compared with proximal nerve constriction.

### 4.2. Differential Pathological Changes after Constriction Distal or Proximal to DRG

These 2 rat models of LR presented differential pathophysiological changes of DRG. Hypoxia is the major pathological mechanism in proximal nerve constriction, whereas neuron injury is the major pathological mechanism in distal nerve constriction. It revealed a more complicated picture of LR and illustrated that there are several different mechanisms for pain chronicity associated with LR. LR includes multiple pain mechanisms caused by mechanical stress, ischemia, inflammation, and nerve damage [15]. Those various pain mechanisms may play different roles in different anatomical locations of nerve injury, and those mechanisms could be revealed by highlighting the difference between animals with nerve constriction and sham animals. The pathological changes of DRG were quite different between these 2 rat models of LR, and the increased ratios of injured CGRP-positive DRG neurons were the only phenomena they shared in proximal nerve constriction, the pathological changes of DRG characterized by hypoxia and increased ratio of CGRP-positive and injured DRG neurons. In contrast, the pathological changes of DRG in distal nerve constriction characterized solely by injured DRG neurons, especially NFH-positive subpopulation. Seeing the longer course of pain behaviors in animals distal nerve constriction, NFH-positive DRG neuron injury may play an essential role in the pain chronicity of LR.

### 4.3. ATF3 Expression Related to Pain

Our study suggested that ATF3 was a bio-signature for the persistent pain in rat models of LR. ATF3, a member of the ATF/CREB family, is rapidly induced in DRG neurons following cellular stress or injury and it is regarded as a reliable marker for neuron injury [30]. The association between ATF3 and neuropathic pain is evident since peripheral nerve injury is the single most reliable predisposing factor for the development of neuropathic pain [31]. However, the mechanisms remain largely unknown. Rau et al. demonstrated that ATF-3 expression was a strong predictor of single cells displaying pain-related electrophysiological properties in a cutaneous tissue damage model [32]. The decrease of ATF-3 expression in DRG neurons by silencing of TNF-α was associated with the amelioration of mechanical allodynia after L5 spinal nerve transection [33]. The ATF3-positive DRG neuron ratio after peripheral nerve injury was also associated with the presence of pain phenotypes [34]. Obata et al. demonstrated that, 2 weeks after the constriction of the common sciatic nerve, animals with the ATF3-positive DRG neuron ratios less than 12.5% did not present mechanical or thermal hyperalgesia [34]. In line with this finding, animals with ATF3-positive DRG neuron ratios less than 12.5% (sham group: 10.74 ± 2.17%) in our study did not present mechanical hyperalgesia and cold allodynia compared to the control group. In contrast, the ATF3-positive DRG neuron ratios were 23.78 ± 2.96 and 39.81 ± 2.41% in the proximal and distal nerve constriction group, respectively, and both groups of rats showed mechanical hyperalgesia and cold allodynia compared to the control group.

### 4.4. The CGRP Expression after Peripheral Nerve Injury

Our study showed that the ratio of CGRP-positive neurons increased 1 day after ligation in proximal nerve constriction but not in distal nerve constriction, but the ratios of injured CGRP-positive DRG neurons increased in both groups, suggesting the universal role of injured CGRP-positive DRG neurons in pain chronicity of LR. CGRP is a 37-amino acid neuropeptide within the calcitonin peptide family and is widely distributed in both the peripheral and the central nervous systems. There was a substantial inconsistency in the expressions of CGRP after peripheral nerve injury in the literature. It seems that the nature of peripheral nerve injury has an impact on CGRP expression dynamics [35]. After sciatic nerve transection, a decrease in CGRP level in DRGs was observed for 45 days [36,37]. However, as opposed to that, a study reported that an increase of CGRP- IR in DRGs after sciatic nerve transection [35]. In the tibial nerve transection model, expression of CGRP was down-regulated 12 days following neurotomy [38]. In the sciatic nerve ligation model, which is associated with a complete blockade of axonal transport, induced a sustained decrease in CGRP-IR in primary sensory neurons of L4-L5 DRGs [35]. Following the sciatic nerve crush model, CGRP-IR was increased in ipsilateral primary sensory neurons of L4-L5 DRGs, and it exhibited a distinct temporospatial pattern, and expression levels had returned to baseline levels by the end of the 28-day test period [35]. In the L5 spinal nerve ligation model, a smaller, sustained decrease occurred in the number of small-, medium- and large-sized neurons immunoreactive for CGRP in the L5 DRG at 1–20 weeks [39,40,41,42]. Our study showed that 1 day after L5 spinal nerve ligation, the ratios of CGRP-IR DRG neurons in L5 DRG significantly increased in proximal spinal nerve injury compared to the sham group. The difference reflected the time dynamics of CGRP expression after spinal nerve ligation.

### 4.5. Clinical Application of Large-Diameter Sensory Afferent Injury in LR

It is critical to prevent acute LR from developing as chronic LR since the current treatments are not effective for chronic LR. Six-month conservative treatment is suggested for acute LR by North American Spine Society [43] because the majority of LR resolved without surgical intervention after 3 months [2]. However, some patients with severe LR developed chronic LR, and the treatment, even surgical intervention, for chronic LR is ineffective [3]. At present, there is not a factor predicting pain chronicity in LR. A nation-wide cohort showed severe LR is associated with pain chronicity [4], and 2 studies suggested that involvement of large-diameter sensory afferents could serve as an index of severity of peripheral nerve injury [5,12]. The present study provided evidence that large-diameter sensory afferent injury contributes to persistent pain in LR. Impaired functions of large-diameter sensory afferents could be easily evaluated by simple sensory tests, which makes it a promising tool to identify patients with severe LR in the acute stage so that treatment will be more aggressive in the acute stage.

Limitations existed in this study. Although we observed the increased ratios of injured NFH-positive DRG neurons after distal nerve constriction, it remains an unanswered question why constriction distal to the DRG results in more injured NFH-positive DRG neurons than does that proximal to the DRG, seeing that the same number of sensory fibers are involved in these two nerve constrictions. Anatomically, the most striking difference is the involvement of motor fibers between these two nerve constrictions. In addition to sensory fibers, motor fiber is involved in distal nerve constriction injury. Substantial evidence showed motor fiber injury alone could lead to neuropathic pain [44] through upregulation of pro-inflammatory cytokines [45,46,47] and ectopic discharge [48]. Considering the larger size of axon and thicker myelin sheath in motor fibers, motor fiber injury in distal nerve constriction may produce a stronger degeneration response and subsequently more ATF3-expression in large-diameter DRG neurons. The increased ratio of ATF3-expressing DRG neurons in distal nerve constriction supported this inference. Another limitation is that we only observed the acute pathological change of DRG after nerve constriction. The DRG may experience some other pathological changes in subacute or chronic stages, which also contribute to pain chronicity of LR.

## 5. Conclusions

This study demonstrated animals with nerve constriction distal to DRG presented more persistent pain behaviors compared with those with nerve constriction proximal to DRG, and the more persistent pain phenotypes are associated with more NFH-positive DRG neuron injury in the DRGs 1 day after nerve injury. Hypoxia was the major pathological change in proximal nerve constriction, whereas nerve injury was the major pathological change in distal nerve constriction. Our results suggest that DRG neuron injury, specifically NFH-positive subpopulations, are the prognostic factors for persistent pain associated with LR at the acute stage.

## Figures and Tables

**Figure 1 cells-10-00992-f001:**
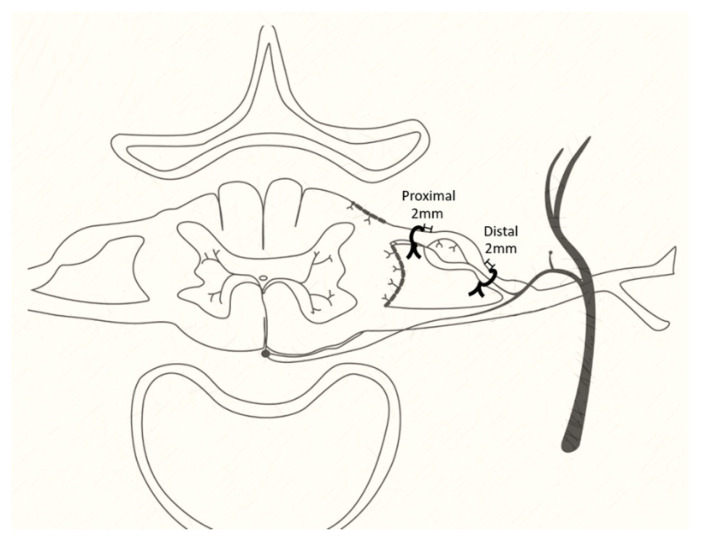
The animal models of lumbar radiculopathy. The 4-0 silk constrictions were applied 2 mm distal to the dorsal root ganglion of the L5 nerve root in animals with distal nerve constriction and 2 mm proximal in those with proximal nerve constriction.

**Figure 2 cells-10-00992-f002:**
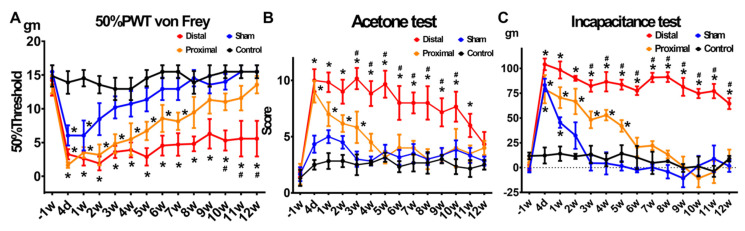
Pain behaviors in animal models of lumbar radiculopathy. (**A**) 50% paw withdraw threshold von Frey test, (**B**) acetone test, and (**C**) weight-bearing test of animals with distal nerve constriction (Distal), proximal nerve constriction (Proximal), sham operation (Sham), and naïve (Control). Data are mean ± SEM, N = 6 rats in each group. * *p* < 0.05 vs. Control, # *p* < 0.05 vs. Proximal, scale bar = 50 μm.

**Figure 3 cells-10-00992-f003:**
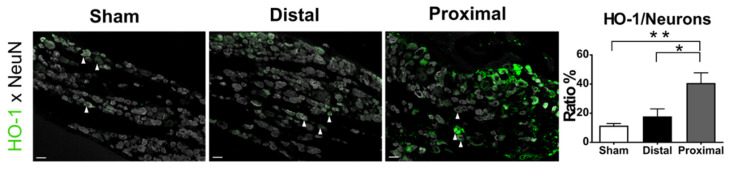
Tissue hypoxia of L5 DRGs at post-operative day 1. Tissue hypoxia was examined in rats labelled with hypoxiaprobe-1 (HO-1, arrowheads), and the L5 DRG sections were stained with FITC-conjugated antibodies recognizing pimonidazole-protein adducts. The ratios of HO-1 positive neurons in L5 DRGs were compared between distal nerve constriction (Distal), proximal nerve constriction (Proximal), and sham operation (Sham) groups. Data are mean ± SEM, N = 6 rats in each group. * *p* < 0.05, ** *p* < 0.01 between groups, scale bar = 50 μm.

**Figure 4 cells-10-00992-f004:**
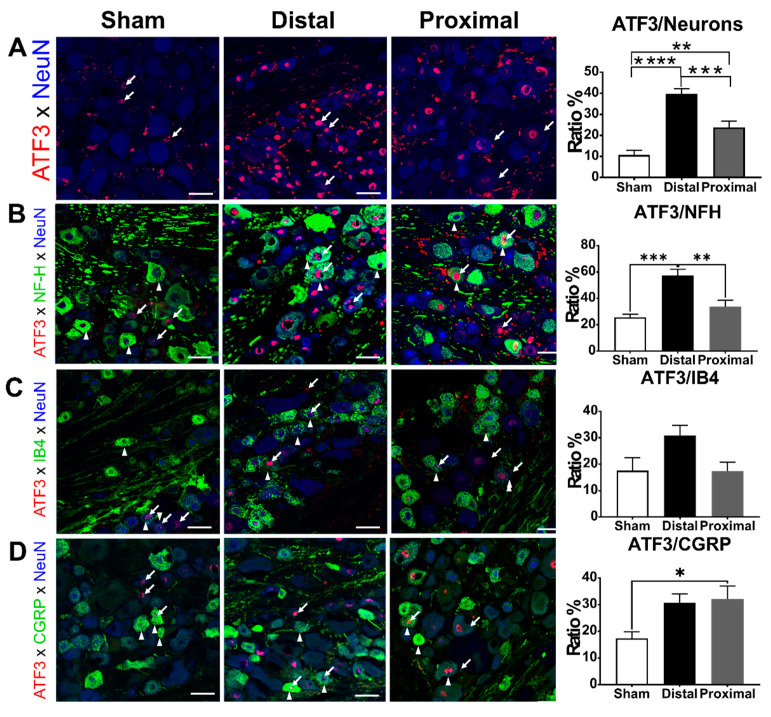
The characterization of ATF3-positive DRG neurons. The ratios of (**A**) total ATF3, (**B**) ATF3/NFH, (**C**) ATF3/IB4, or (**D**) ATF3/CGRP colocalized DRG neurons of animals with distal nerve constriction (Distal), proximal nerve constriction (Proximal), or sham operation (Sham). Data are mean ± SEM, N = 6 rats in each group. * *p* < 0.05, ** *p* < 0.01, *** *p* < 0.001, **** *p* < 0.0001, scale bar = 50 μm. Arrows indicate the ATF3 expression, arrowheads indicate a neuron subtype marker (NFH, IB4, or CGRP).

**Figure 5 cells-10-00992-f005:**
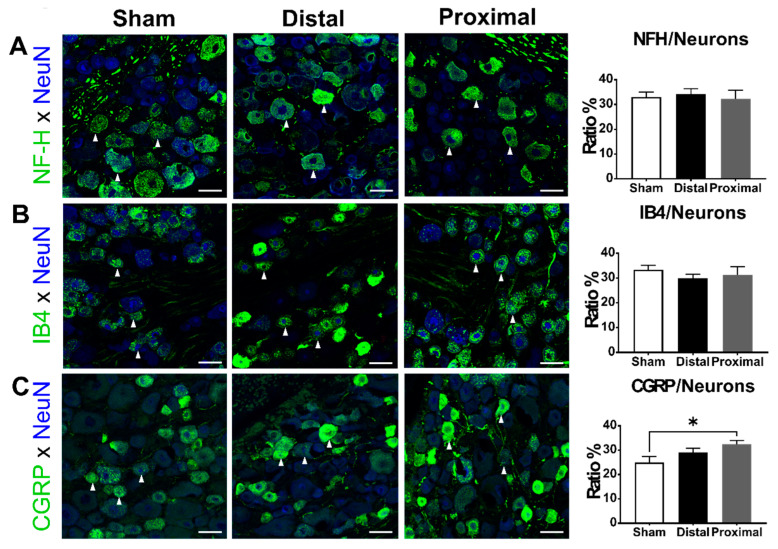
The subpopulation of DRG neurons 1 day after the operation. The ratios of (**A**) NFH, (**B**) IB4, or (**C**) CGRP-positive (arrowheads) DRG neurons of animals with distal nerve constriction (Distal), proximal nerve constriction (Proximal), or sham operation (Sham). Data are mean ± SEM, N = 6 in each group. * *p* < 0.05, scale bar = 50 μm.

## Data Availability

In this section, please provide details regarding where data supporting reported results can be found, including links to publicly archived datasets analyzed or generated during the study. Please refer to suggested Data Availability Statements in section “MDPI Research Data Policies” at https://www.mdpi.com/ethics. You might choose to exclude this statement if the study did not report any data.

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
