# Peer review of "ATF3-Expressing Large-Diameter Sensory Afferents at Acute Stage as Bio-Signatures of Persistent Pain Associated with Lumbar Radiculopathy"

_cells, 2021, doi:10.3390/cells10050992_

Round 1
Reviewer 1 Report
In this manuscript, the authors demonstrated that animals with nerve constriction distal to DRG have more severe and prolonged pain behaviors compared to the injury proximal to DRG, and suggested that the ATF3 expression in NFH-positive DRG neurons may contribute to the observed differences. The manuscript is well written and data is clearly presented. However, the conclusions are not convinced due to the limitation of the experimental design.
- Given all the immunohistochemistry data were obtained at 1 day post-injury, it is hard to believe the pathological changes at the 1st 24 hours can determine the long-term consequences. It is important to examine whether the similar experimental findings can also be observed at late time points.
- In the behavioral data presented, are all the results derived from the measurements in the ipsilateral side? The formula in the subtitle “Incapacitance test” on page 4 should reflect the % weight distribution of the right hind paw, not the left hind paw; and in the “Acetone test”, it is unclear whether the scores are summed from one side or both sides of the paws.
- Is there an increased expression of CGRP in NFH positive neurons?
- It might be more informative to see whether the distal and proximal injuries to DRG are associated with different inflammatory responses.
- The ATF3 images in some panels in Figure 4 are less clear.
- The increased expression of ATF3 in larger diameters, but not in the smaller diameters of neurons should be discussed.
Author Response
- In this manuscript, the authors demonstrated that animals with nerve constriction distal to DRG have more severe and prolonged pain behaviors compared to the injury proximal to DRG, and suggested that the ATF3 expression in NFH-positive DRG neurons may contribute to the observed differences. The manuscript is well written and data is clearly presented. However, the conclusions are not convinced due to the limitation of the experimental design.
Response:
We had modified the conclusion and title as ATF3-expressing of large diameter neurons at acute stage associated with chronic pain in lumbar radiculopathy.
- Given all the immunohistochemistry data were obtained at 1 day post-injury, it is hard to believe the pathological changes at the 1st 24 hours can determine the long-term consequences. It is important to examine whether the similar experimental findings can also be observed at late time points.
Response:
Thank you for this valuable comment.
We agree with you and fully understand that long-term consequence may not be determined by the acute pathological changes. However, we also can not deny the possibility that acute pathological changes could play an important role in chronic consequence. In this present study, we focus only on the acute pathological changes and not on chronic changes. Besides, the results of this only established the association between acute pathological changes and long-term consequences, instead of causal effect. For clearance, we modify the conclusion as ATF3-expressing of large diameter neurons at acute stage associated with chronic pain in lumbar radiculopathy.
- In the behavioral data presented, are all the results derived from the measurements in the ipsilateral side? The formula in the subtitle “Incapacitance test” on page 4 should reflect the % weight distribution of the right hind paw, not the left hind paw; and in the “Acetone test”, it is unclear whether the scores are summed from one side or both sides of the paws.
Response:
Thank you for the comment.
Only the scores of the right paw are summed in acetone test. For clearance, the description was modified. Please see page 6.
- Is there an increased expression of CGRP in NFH positive neurons?
Response:
Thank you for valuable comments.
This is an interesting point. Unfortunately, we did not see the colocalization of CGRP and NFH immunoreactivity in this present study.
- It might be more informative to see whether the distal and proximal injuries to DRG are associated with different inflammatory responses.
Response:
Thank you for valuable comments.
We will consider inflammatory responses in next study.
- The ATF3 images in some panels in Figure 4 are less clear.
Response:
Thank you.
The ATF3 images have been improved.
- The increased expression of ATF3 in larger diameters, but not in the smaller diameters of neurons should be discussed.
Response:
Thank you for comments.
We have added a discussion regarding this issue in the discussion section. Please see page 16.
Reviewer 2 Report
It is a very interesting study. Thank you so much. In particular, the comparison between distal and proximal constriction surgery is really interesting point.
- Major concerns
- Thermal pain has been reported in many previous studies. You checked the cold allodynia. Did you check the Hargreave's test?
2. You presented the IHC quantification data in Fig 4, 5. However, usually, many researcher do not use IHC data to quantify. It is also true that there are problems. The IHC data should present the co-localization, and it would be better to present the difference in protein or mRNA expression levels by western blot or qPCR data for quantification.
- Minor concern
- You identified the HO-1 as an inflammation reaction, but did you check any other markers?
- Did you check the EPSC of ATF3 positive DRG neurons? If possible, please add electrophysiology data. (Patch clamp data or calcium imaging data of population).
Author Response
- It is a very interesting study. Thank you so much. In particular, the comparison between distal and proximal constriction surgery is really interesting point.
Response: Thank you for encouragement!
- Thermal pain has been reported in many previous studies. You checked the cold allodynia. Did you check the Hargreave's test?
Response:
Thank you.
We did not check Hargreave’s test.
- You presented the IHC quantification data in Fig 4, 5. However, usually, many researcher do not use IHC data to quantify. It is also true that there are problems. The IHC data should present the co-localization, and it would be better to present the difference in protein or mRNA expression levels by western blot or qPCR data for quantification.
Response:
ATF3 expression of dorsal root ganglion (DRG) was not different among the proximal group and the distal group and sham group by Western blot, although the IHC showed elevated ATF3-expression of DRG neurons in the distal group.
It could be explained by that non-neuronal cells contribute the majority of ATF3 expression in DRG after injury. (J Immunol. 2015 Nov 1;195(9):4446-55.). DRG neurons only contribute the minority of ATF3 expression. However, the focus of the present study is the pathological changes of DRG neurons, because the pathological changes of DRG neurons is more likely to be associated with chronic pain.
- You identified the HO-1 as an inflammation reaction, but did you check any other markers?
Response:
Thank you for comments. HO-1 is a marker for tissue hypoxia. Thereafter, we regard it as a hypoxia marker. We did not check other inflammation markers. We will consider including inflammation reaction in the next study.
- Did you check the EPSC of ATF3 positive DRG neurons? If possible, please add electrophysiology data. (Patch clamp data or calcium imaging data of population).
Response:
Thank you for valuable comments.
It would be very interesting to investigate the functional activity, such as EPSC or calcium image, of ATF3 -positive DRG neurons and see their relationship with pain chronicity. More than this, other ATF3-negative DRG neurons should be investigated as well since they may also play an important role in pain chronicity. Unfortunately, we did not perform the functional activities in this present study. We will consider including this in the next study. Thank you again.

Round 2
Reviewer 1 Report
The authors have reasonably addressed my concerns, and the quality of the manuscript is now improved.
Reviewer 2 Report
Dear authors,
Thank you for kindly answering my concerns. Since I am an electrophysiologist, I am a bit disappointed that functional studies are missing from your study. But it's a really interesting study.
I always hope for good your research.
Best,
Han